# Affordances for Motor Development in the Home Environment for Young Children with and without CHARGE Syndrome

**DOI:** 10.3390/ijerph182211936

**Published:** 2021-11-13

**Authors:** Pamela Beach, Melanie Perreault, Lauren Lieberman

**Affiliations:** Department of Kinesiology, Physical Education, Sport Studies, State University of New York Brockport, Brockport, NY 14420, USA; mperreault@brockport.edu (M.P.); llieberman@brockport.edu (L.L.)

**Keywords:** motor development, disability, deafblindness, motor skills

## Abstract

Affordances in the home environment are critical to early motor development. Currently, the home environment has not been examined in children with deafblindness or severe disabilities. The present study examined differences in, and relationships between, the home environment and motor development in children with and without CHARGE syndrome. CHARGE syndrome is a low-incidence, complex disorder with sensory and motor impairments. Participants included 28 parents of children with CHARGE syndrome and 32 parents of children without disabilities. Children with CHARGE syndrome achieved motor milestones significantly later and had fewer outside space affordances than children without disabilities. Older children had a greater variety of stimulation and fine motor toys, and those that achieved independent walking later had more outside space and fine and gross motor toys. Early experiences may be more important for children with CHARGE syndrome than children without disabilities. Moreover, parents can play a vital role in their children’s motor development to help them reach their motor milestones.

## 1. Introduction

During the early years, young children develop many motor milestones and fundamental motor skills [1]. The home environment is an important source of support for promoting motor development during this critical time of early childhood when a child is provided with a stimulating and supportive surrounding [2]. Infants and young children develop these motor milestones and motor skills through interacting with the environment in a meaningful way. Both gross motor skills, involving large muscles (e.g., kicking), and fine motor skills, involving smaller muscles (e.g., building a tower), are functionally integrated during development [3] and develop using the same higher-order neuromotor processes [4]. These motor skills are the building blocks for more advanced skills that are developed throughout childhood.

The notion of affordances within the ecological perspective provides a framework for understanding external influences on an individual’s motor development. From this perspective, affordances are aspects of the task (e.g., equipment) and environment (e.g., open space, parental support) that provide opportunities for action [5,6]. A supportive home environment for promoting motor development would include family members who are actively involved with the child as well as large spaces (e.g., backyard) and a variety of toys (e.g., balls) that encourage movement. Children with increased support and opportunities in the home would benefit while children who lack these affordances may lag behind their peers in their motor development [7].

Using the ecological perspective, Rodrigues and colleagues [7] created the first inventory addressing the presence of affordances in the home and the effect of these affordances upon motor development titled Affordances in the Home Environment for Motor Development Self-Report (AHEMD-SR). A positive relationship has been found among the five factors of the AHEMD-SR (outside space, inside space, variety of stimulation, gross motor toys, and fine motor toys) and motor development, with the greatest predictor being the child’s fine motor toys [8]. Moreover, children’s access to fine and gross motor toys and parents’ level of physical activity involvement with their children have been shown to significantly impact their motor development [9].

The AHEMD has also been used to examine differences in the home environment in infants and young children with and without disabilities, such as sensory impairments. Lage and colleagues [10] found that children with low vision had significantly less fine and gross motor toy affordances than their sighted peers; however, there were no differences in caregiver assistance between the two samples. Additionally, Araujo and colleagues [11] found that homes of infants with risk indicators for hearing loss displayed statistically fewer affordances when compared to homes of infants without the risk indicators. These findings revealed a variety of factors that influenced the development of fine and gross motor skills at home including toy availability, parental physical activity involvement, parental socioeconomic status (SES), age, and level of children’s hearing [11]. Although studies have been conducted upon young children with visual and hearing losses separately, the influence of the affordances for motor development in the home environment has not been examined in young children with both visual and hearing losses or severe disabilities. To examine the importance of the home environment upon motor development in young children with deafblindness and severe disabilities, this study sought to examine young children with CHARGE syndrome, which is the leading genetic cause of congenital deafblindness [12]. 

CHARGE syndrome is a rare genetic disorder identified by a broad range of multisensory deficits. One identifying factor is a mutation to the *CHD7* gene, which has been identified as the cause of CHARGE syndrome [13]. A diagnosis of CHARGE syndrome is based on testing for the presence of a *CHD7* gene mutation and/or a combination of the major and minor features [14]. Currently, the selection of these factors and the combination of factors needed for diagnosis are still under consideration. Major features currently include coloboma of the eye, choanal atresia, cranial nerve anomalies, and ear malformation. Minor features currently include heart defects, cleft lip and palate, kidney abnormalities, genital abnormalities, growth deficiency, typical CHARGE face, and palmar crease [14]. 

Children with CHARGE syndrome signify a heterogeneous population; therefore, individuals vary widely in their lived experiences of the medical and physical difficulties associated with the syndrome [15]. However, most individuals with CHARGE syndrome experience some level of hearing loss, vision loss, and balance problems, all of which lead to various levels of delays in motor development and communication [16]. In addition to physical limitations, children with CHARGE syndrome often experience multiple, prolonged hospitalizations, which may increase the likelihood of developmental delays due to their lack of consistent socialization and physical activity [17]. Based on the challenges described above, it is not surprising that children with CHARGE syndrome experience delays in the achievement of motor milestones [18]. Motor milestones are early motor skills that follow a particular sequence from holding the head up to independent walking. Children with CHARGE syndrome walk an average of 25 months later than their sighted peers [19,20]. One reason for this late onset for independent walking is that children with CHARGE syndrome experience more balance issues compared to their peers without disabilities [21,22]. This delayed walking can have drastic impacts on other gross motor skills, such as locomotor and object control skills [23,24], which are critical for engaging in many physical activities. 

To date, it is not known if motor delays have been affected by the home environment. Although specific motor development characteristics have been assessed for children with CHARGE syndrome, the influence of the home environment upon their motor development has not been examined. The first aim of this study was to use the AHEMD to examine differences in affordances for motor development in the home environment between young children with and without CHARGE syndrome to better understand the multidimensional effects of affordances in the home related to motor development upon disability. A second aim of this study was to compare the relationship between home environment affordances with the age of onset of motor milestones in young children with CHARGE syndrome. To examine these aims, we proposed two hypotheses: 1. affordances as measured by the AHEMD will significantly differ in children with CHARGE syndrome in comparison to children without disabilities; 2. fine and gross motor toys available in the home will be significantly related to age of motor milestones.

## 2. Materials and Methods

### 2.1. Participants

Participants included parents or legal guardians of a child with or without CHARGE syndrome between the ages of 18 and 42 months. Twenty-eight parents of children with CHARGE syndrome and 32 parents of children without disabilities participated in this study. The children with CHARGE syndrome included 14 females and 14 males; 20 Caucasian, 1 American Indian or Alaskan Native, 2 Asian, 1 other, 3 more than one ethnicity, and 1 not indicated. Children without disabilities included 15 females and 17 males; 21 Caucasian, 2 Black or African American, 2 other, 4 more than one ethnicity, and 5 not indicated. Refer to Table 1 for descriptive data on the children of the participants and Table 2 for the CHARGE characteristics of the children with CHARGE syndrome. Independent-samples *t*-tests and chi-square analyses revealed no significant differences in age (t (5) = 0.40, *p* = 0.70), height (t (54) = −0.52, *p* = 0.61), gender (*X*^2^ (1, *N* = 60) = 0.06, *p* = 0.81), or ethnicity (*X*^2^ (4, *N* = 54) = 3.5, *p* = 0.48) between the children of the two samples; however, there was a significant difference in weight (t (57) = −2.86, *p* = 0.006), wherein the children with CHARGE syndrome weighed significantly less than the children without disabilities. This is not surprising because growth is often restricted in individuals with CHARGE syndrome [14]. 

### 2.2. Measure

Affordances in the Home Environment for Motor Development is a self-report instrument that assesses the quality and quantity of motor development opportunities in the home during early childhood [7,25]. The measure is designed for children between the ages of 18 and 42 months old and completed by parents or legal guardians, which are the best source of information on the home environment. The AHEMD inventory consists of 67 questions examining the home and characteristics of the family. The inventory consists of five sections: two on the family and child’s characteristics and three on affordances of the home environment: (1) physical space, (2) daily activities, and (3) play materials. There are three types of questions including Likert scale, multiple choice, and descriptive queries. The scoring method specifically designed for this inventory was used in this study. Total scores range from 5 to 20. Scores below 10 are categorized as low, meaning minimal affordances to promote motor development, scores between 10 and 15 are categorized as average, indicating sufficient opportunities, and scores between 15 and 20 are categorized as high, meaning very good affordances in the home for motor development. In addition, the survey includes a short family demographic survey consisting of the following variables: number of adults living in the house, number of children living in the house, number of rooms in the house, length of time the child has lived in the house, father’s education, mother’s education, annual family income, and length of childcare attendance. All demographic items are scored on a Likert scale. In addition, parents indicated the age of onset for their child’s motor milestones.

Construct validity and reliability were previously examined with families from the United States and Portugal [7]. Confirmatory factor analysis provided a 5-factor solution: outside space, inside space, variety of stimulation, fine motor toys, and gross motor toys. Fit indexes were all over 0.90 and reliability was established with high consistency, construct of interest ranging from 0.80 to 0.91. 

### 2.3. Procedure

Parents of children with CHARGE syndrome were recruited from the CHARGE Syndrome Foundation via email and social media, and parents of children without disabilities were recruited from local daycare facilities (letters) and social media. Participants completed the survey by accessing a link or QR code that included the informed consent, demographics, CHARGE syndrome characteristics, age of onset of motor milestones, and AHEMD. Institutional review board approval was received prior to recruiting for this study.

### 2.4. Analyses

Scores from each participant on the AHEMD were entered into the AHEMD calculator provided by the developers to obtain standard scores for the five different factors: (1) outside space, (2) inside space, (3) variety of stimulation, (4) fine motor toys, and (5) gross motor toys [25]. Standard scores range from 1 (very low) to 4 (high). These scores were then combined to form a total AHEMD score ranging from 5 to 20. 

A chi-square analysis and independent-samples *t*-tests were conducted to examine differences in the achievement and age of motor milestone between the children with CHARGE syndrome and the children without disabilities. Next, Mann–Whitney U tests were performed to determine differences in family demographic variables and the standard scores of the AHEMD between the two samples. Finally, Spearman rank order correlations were used to examine relationships between current age, age of each motor milestone, family demographics, and AHEMD scores for each sample. 

## 3. Results

### 3.1. Motor Milestones

The chi-square analysis revealed significant differences in the achievement of several motor milestones between the two samples. Specifically, more children without disabilities could crawl, creep, stand without support, cruise, walk, run, jump, and hop than the children with CHARGE syndrome (see Table 3). There were also significant differences in age of motor milestones, wherein the children without disabilities achieved each milestone at an earlier age than the children with CHARGE syndrome (*p* < 0.001; see Table 4). Due to the limited number of children with CHARGE syndrome who could run, jump, and hop, these motor milestones were excluded from the analysis. Family demographic variables only differed significantly between the two samples for childcare length (U = 196, *p* < 0.001, *r* = 0.45), wherein the children without disabilities had longer childcare attendance than the children with CHARGE syndrome.

### 3.2. AHEMD Scores

Frequencies of affordance level for AHEMD scores across the two samples are presented in Table 5. Overall, most children across the two samples were categorized in the good to high range across all components of the AHEMD. One notable difference was that no children without disabilities were categorized as extremely low for any component whereas a small number of children with CHARGE syndrome were categorized as very low for outside space and variety of stimulation. Interestingly, all participants were categorized as high for inside space; thus, this component does not appear to be a discriminating factor between the two samples. For the total AHEMD score, all participants were categorized as average or high; however, a much larger majority of the children without disabilities were categorized as high compared to the children with CHARGE syndrome. Despite some of these initial observations, the analysis revealed that the two samples only differed on outside space (U = 299, *p* = 0.05, *r* = 0.25), suggesting that parents of children without disabilities offer better outside space affordances than the parents of children with CHARGE syndrome (see Table 6).

For the children with CHARGE syndrome, the correlation analysis revealed that current age had a significant positive relationship with variety of stimulation (*r* = 0.41, *p* = 0.039) and fine motor toys (*r* = 0.50, *p* = 0.01), indicating that the older the child, the more variety of movement opportunities and fine motor toys the child had access to in the home. The analysis also indicated that age of standing without support had a significant positive correlation with fine motor toys (*r*_s_ = 0.60, *p* = 0.038) and total AHEMD (*r*_s_ = 0.64, *p* = 0.024). Age of walking had significant positive relationships with outside space (*r*_s_ = 0.63, *p* = 0.038), fine motor toys (*r*_s_ = 0.70, *p* = 0.016), gross motor toys (*r*_s_ = 0.84, *p* = 0.001), and total AHEMD (*r*_s_ = 0.88, *p <* 0.001). However, these results must be interpreted cautiously due to the small number of children who had reached these milestones, *n* = 12 and *n* = 11, respectively. For the demographic variables, number of adults in the house had significant negative correlations with outside space (*r*_s_ = −0.40, *p* = 0.041) and gross motor toys (*r*_s_ = −0.42, *p* = 0.032), indicating that children had less access to outside space and gross motor toys when there were more adults living at the family house. Significant positive relationships were found for number of rooms in the house with both outside space (*r*_s_ = 0.46, *p* = 0.019) and total AHEMD (*r*_s_ = 0.43, *p* = 0.029); number of children in the house correlated with age of walking (*r*_s_ = 0.66, *p* = 0.026), fine motor toys (*r*_s_ = 0.47, *p* = 0.017) and total AHEMD (*r*_s_ = 0.47, *p* = 0.019); and annual income correlated with age of walking (*r*_s_ = 0.68, *p* = 0.022), gross motor toys (*r*_s_ = 0.46, *p* = 0.017) and total AHEMD (*r*_s_ = 0.51, *p* = 0.008).

For the children without disabilities, there were no significant relationships between current age and age of motor milestones with the AHEMD. However, there were significant correlations for some of the demographic variables. Specifically, number of children in the house had significant positive relationships with outside space (*r*_s_ = 0.39, *p* = 0.03), variety of stimulation (*r*_s_ = 0.51, *p* = 0.004), and total AHEMD (*r*_s_ = 0.43, *p* = 0.018), indicating that outside space and varied movement opportunities were more available when more children were living in the home. There was also a significant positive association between annual income and variety of stimulation (*r*_s_ = 0.41, *p* = 0.024), which suggests that wealthier families were able to provide more diverse movement opportunities at home.

## 4. Discussion

Motor skill development has an interactive effect with cognitive, emotional, and motor-perceptual development [8]. The purpose of this study was to examine the affordances in the home environment in young children with and without CHARGE syndrome to examine the multidimensional effects of affordances to motor development. Home environment resources include toys and space, but also parental and family support such as encouragement, guidance, and regular engagement with their child. To the author’s knowledge, this is the first study to examine the home environment in young children with deafblindness or severe disabilities. Specifically, we examined young children with CHARGE syndrome, the leading cause of congenital deafblindness [12]. Motor development is often significantly delayed in children with CHARGE syndrome due not only to sensory impairments, but motor impairments as well as environmental constraints, such as long hospital stays [23,24]. 

To examine the affordances for motor development in the home environment, parents of young children with and without CHARGE syndrome completed a validated survey on affordances in the home environment, AHEMD [7,25]. The rationale for this research is that more affordances in the home afford children more opportunities for movement experiences and physical activity that can promote fine and gross motor development [26]. This notion is based upon the ecological theory in which environments have individuals, objects, places, and events that provide action opportunities [6]. These action opportunities, affordances, are experienced differently depending upon the child’s functionalities and capabilities [6,27], and therefore, would be expected to be different in children with deafblindness and severe disabilities in comparison to children without disabilities.

Due to the delays in motor development, it was important to examine the role of demographics of the family and affordances in the home environment with the achievement of motor milestones in children with and without CHARGE syndrome. The findings revealed that children without disabilities were in childcare significantly more than children with CHARGE syndrome. These results are likely due to the medical needs of children with CHARGE syndrome at an early age [23,24]. Most daycares are typically not equipped to attend to the additional medical, functional, and communication needs of children with CHARGE syndrome. Outside space was also found to be higher for children without disabilities in comparison to the children with CHARGE syndrome. Although not as important as inside space, outdoor space is an important affordance contributing to motor development [28]. Outside space included more than one ground texture, one or more sloped surfaces, and apparatus to grasp, climb, or step, such as a playground. Limited space is a predictor of gross motor development in children without disabilities [28] and would likely restrict development in children with disabilities to an equal or potentially greater extent than children without disabilities. It is possible that parents of children with CHARGE syndrome purchase less outdoor equipment as a response to their child’s motor delays assuming that their child is not developmentally ready to use the equipment. It is also possible that parents perceive outdoor equipment as more dangerous than indoor toys. Parents of children with visual impairments are often found to be overprotective out of fear for their child’s safety [29].

The findings of this study also provided some implications of the family demographics as affordances to their child’s motor development. Interestingly, children with CHARGE syndrome lived in families with more adults than the families of children without disabilities. This may be due to the additional assistance and the presence of significant stress found in families with a non-verbal child [30]. Other demographic factors that influenced the affordances for children with CHARGE syndrome were SES and number of siblings. SES, which includes income and parental education, has been associated with the level of home affordances, wherein children who live in homes with higher incomes and parental education are provided with more affordances leading to higher motor skill development [28,31]. In the present study, SES, specifically income, was associated with age of walking, gross motor toys, and the total AHEMD score for children with CHARGE syndrome and SES was associated with a variety of stimulation for children without disabilities. 

A second aim of this study was to examine the relationship between home environment affordances with the age of onset of motor milestones in young children with CHARGE syndrome due to their delays in motor development [32]. Similar to prior research [18,19,20], the results of the present study found significant delays in the acquisition of motor milestones in the children with CHARGE syndrome, such as age of standing and independent walking occurring significantly later than their peers without disabilities. Interestingly, however, age of walking was associated with the number of toys in the home. It is possible this is due to parents purchasing more toys to encourage their child to walk if they are delayed. Children who develop gross motor skills early likely would not need as many toys to promote walking. On another note, age was also positively associated with a variety of stimulation and fine motor toys in the present study for children with CHARGE syndrome. Older children benefitted from more fine motor toys, such as peg boards or lacing cubes, and more opportunities for movement than younger children. It is possible that there are more toys for older children due to the parents having more time to purchase toys and parents are more likely to purchase toys that they feel may benefit their child. Regardless of the reason for increased toys, it is important to note that fine motor toys are a significant predictor of fine motor skill development [28]. 

### Limitations

This study provides some initial findings of affordances for motor development in the home environment in an underrepresented population, children with severe disabilities, specifically, CHARGE syndrome. Due to the low incidence of this population, obtaining large sample sizes, which is more typical of this type of research, is not possible. It is important to also understand the uniqueness of CHARGE syndrome, which can reduce generalizability. However, considering there has not been research conducted with children with severe disabilities, it is possible the findings may generalize to families with children with severe disabilities as they may likely have many similar challenges at home. As such, more research should be conducted on other populations with severe disabilities to strengthen these results and increase knowledge of the home affordances for children with severe disabilities. 

It is also important to note that due to the delayed gross motor development of children with CHARGE syndrome, there was a particularly small sample size of participants who were able to independently walk. Children with CHARGE syndrome typically walk independently 30 months later than their peers without disabilities [22]. As such, future research should also examine broader age groups in families with children with severe disabilities. 

Furthermore, it is important to mention the limitations of using a parental questionnaire. The questionnaire we used has been found to be valid and reliable; however, a thorough home assessment would provide more depth than parental report. For example, the assessment does not examine the length of time children spend using the affordances. The current research also relied upon parental recall for the onset of motor milestones. The researchers feel that although the assessment relied upon parental assessments and recall, the findings provide a foundation for understanding home affordances in children with severe disabilities, which, as mentioned, should be further explored. Future research should extend upon these findings by also examining the relationship of home affordances with objective measurements of fine and gross motor skills. 

## 5. Conclusions

The home environment provides important affordances that can either promote or inhibit fine and gross motor development. The findings of this study indicate that early experiences may be even more important for children with CHARGE syndrome in comparison to children without disabilities. Parents should help guide their children with CHARGE syndrome on their motor development. Home environment resources include toys and space, but also parental and family support such as encouragement, guidance, and regular engagement with their child. In addition, it is recommended for parents of children with CHARGE syndrome to work with their child’s support team to determine optimal home affordances.

## Figures and Tables

**Table 1 ijerph-18-11936-t001:** Descriptive data of children with and without CHARGE syndrome.

	Children with CHARGE		Children without Disability	
	Mean	SD	Mean	SD
Children’s Age	29 mths	9.6 mths	28.1 mths	7.5 mths
Weight	25.1 lbs	4.0 lbs	28.8 lbs	5.2 lbs
Height	33.9 in	9.8 in	34.9 in	2.8 in

**Table 2 ijerph-18-11936-t002:** CHARGE characteristics.

Visual	17	Ocular Colobomas	14	Loss of Visual Field	0	CVI								
Visual Acuity (right)	15	20/199 or better	6	20/199 to 599	1	20/600 and up								
Visual Acuity (left)	13	20/199 or better	6	20/199 to 600	2	20/600 and up								
Hearing Loss (right)	2	Normal (−10–15)	5	Slight (16–25)	3	Mild (26–40)	0	Moderate (41–55)	3	Moderate Severe (56–70)	1	Severe (71–90)	13	Profound (90+)
Hearing Loss (left)	2	Normal (−10–15)	2	Slight (16–25)	0	Mild (26–40)	1	Moderate (41–55)	2	Moderate Severe (56–70)	4	Severe (71–90)	17	Profound (90+)
Other Characteristics	20	Heart defects	11	Atresia of Choanae	19	Growth restrictions			26	Missing or Malformed Semicircular Canals				

**Table 3 ijerph-18-11936-t003:** Cross tabulation of children with and without CHARGE syndrome and achievement of motor milestones.

Motor Milestone		CHARGE (*N* = 28)	Control (*N* = 32)	*X^2^*	*p*
Holding head	Yes	27	32	1.16	0.28
	No	1	0		
Rolling over	Yes	28	32		
	No	0	0		
Sitting w/o support	Yes	25	32	3.61	0.06
	No	3	0		
Crawling	Yes	20	30	6.25	0.01
	No	7	1		
Creeping	Yes	20	32	10.55	0.001
	No	8	0		
Standing with support	Yes	26	31	2.29	0.13
	No	2	0		
Standing w/o support	Yes	13	31	22.27	<0.001
	No	15	0		
Cruising	Yes	23	31	4.93	0.03
	No	4	0		
Walking	Yes	12	31	24.31	<0.001
	No	16	0		
Running	Yes	7	29	34.44	<0.001
	No	21	0		
Jumping	Yes	5	23	21.53	<0.001
	No	23	6		
Hopping	Yes	1	14	14.68	<0.001
	No	27	15		

**Table 4 ijerph-18-11936-t004:** Results of *t*-tests and means and standard deviations for age of motor milestone between children with and without CHARGE syndrome.

Motor Milestone	CHARGE (*N* = 28)		Controls (*N* = 32)		t	df	*p*	*d*
	Mean	SD	Mean	SD				
Holding head	8.68	5.01	2.32	1.42	5.29	20.81	<0.001	1.73
Rolling over	10.05	3.40	3.87	2.00	6.21	28.43	<0.001	2.22
Sitting w/o support	13.83	7.41	6.13	1.72	4.30	18.72	<0.001	1.43
Crawling	16.19	5.85	7.61	2.56	5.42	20.05	<0.001	1.90
Creeping	19.54	7.71	8.47	2.44	5.01	13.65	<0.001	1.94
Standing with support	19.32	8.47	8.58	2.51	5.30	21.13	<0.001	1.72
Standing w/o support	23.50	6.19	10.58	3.10	6.73	14.50	<0.001	2.64
Cruising	22.39	10.21	10.76	2.46	4.71	18.87	<0.001	1.57
Walking	26.27	6.44	12.42	2.34	6.88	11.55	<0.001	2.86

**Table 5 ijerph-18-11936-t005:** Frequencies of affordance level for AHEMD scores for children with and without CHARGE syndrome.

Variable	CHARGE (*N* = 28)				Controls (*N* = 32)			
	Very Low	Low	Good or Average	High	Very Low	Low	Good or Average	High
Outside Space	2 (7.7%)	6 (23.1%)	4 (15.4%)	14 (53.8%)	0	2 (6.5%)	6 (19.4%)	23 (74.2%)
Inside Space	0	0	0	26 (100%)	0	0	0	31 (100%)
Variety of Stimulation	1 (3.8%)	4 (15.4%)	3 (11.5%)	18 (69.2%)	0	0	7 (22.6%)	24 (77.4%)
Fine Motor Toys	0	4 (15.4%)	14 (53.8%)	8 (30.8%)	0	1 (3.2%)	19 (61.3%)	11 (35.5%)
Gross Motor Toys	0	0	11 (42.3%)	15 (57.7%)	0	1 (3.2%)	7 (22.6%)	23 (74.2%)
Total AHEMD		0	10 (38.5%)	16 (61.5%)		0	5 (16.1%)	26 (83.9%)

**Table 6 ijerph-18-11936-t006:** Comparison of AHEMD scores between children with and without CHARGE syndrome.

	CHARGE (*N* = 28)		Controls (*N* = 32)		*z*	*p*	*r*
	Median	Mrank	Median	Mrank			
Outside Space	4	25.00	4	32.35	1.97	0.05	0.25
Inside Space	4	29.00	4	29.00	0.00	1.00	0
Variety of Stimulation	4	27.06	4	30.63	1.05	0.29	0.14
Fine Motor Toys	3	27.08	3	30.61	0.91	0.36	0.12
Gross Motor Toys	4	26.65	4	30.97	1.19	0.23	0.15
Total AHEMD	16	24.88	19	32.45	1.75	0.08	0.23

## Data Availability

This study did not report any data.

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
