# Peer review of "Affordances for Motor Development in the Home Environment for Young Children with and without CHARGE Syndrome"

_ijerph, 2021, doi:10.3390/ijerph182211936_

Round 1

Reviewer 1 Report

This paper described the affordances in the home for young children with and without CHARGE syndrome. Unfortunately, I feel that the current manuscript was not sufficient to the required level for publication.  

There are significant grammatical corrections and spelling error that need to be made throughout. For example title.

There is a lack of fidelity in regards to who is taking the methods, study design, discussion point about results etc.

Introduction:

The introduction was a bit long and hard to follow. I know the story you are trying to tell, but I think some of it gets lost.

The first question for the authors is what was the novelty of this study undertaken?

Methods: 2.1 you need to for participants characteristics table.

  • Participants included parents or legal guardians of a child with or without CHARGE 108 syndrome between the ages of 18 and 42 months. Twenty-eight parents of children with     109 CHARGE syndrome (Mage = 29 months, SD = 9.6 months; Mweight = 25.1 pounds, SD = 4.  110 pounds; Mheight = 33.9 inches, SD = 9.8 inches;  14 Females, 14 Males; 20 Caucasian, 1  111 American Indian or Alaskan Native, 2 Asian, 1 other, 3 more than one ethnicity, and 1 not  112 indicated) and 32 parents of children without disabilities (Mage = 28.1 months, SD = 7.5  113 months; Mweight = 28.8 pounds, SD = 5.2 pounds; Mheight = 34.9 inches, SD = 2.8 inches; 15  114 Females, 17 Males; 21 Caucasian, 2 Black or African American, 2 other, 4 more than one  115 ethnicity, and 5 not indicated) participated in this study. Independent samples t-tests and  116 chi-square analyses revealed no significant differences in age (t(5) = .40, p = .70), height  117 (t(54) = -.52, p = .61, gender (X2(1, N = 60) = .06, p = .81), or ethnicity (X2(4, N = 54) = 3.5, p = 118 .48) between the two samples; however, there was a significant difference in weight (t(57)  119 = -2.86, p = .006) ……

Discussion: This is where you need to discuss your results in the context of prior work. What is so important about your findings.

No have Reference 30 in the reference part.

Author Response

Thank you for your comments.  Please find our response in the attached file.

Reviewer 2 Report

Title: Affordances in the Home for Young Children with and without CHARGE Syndrome

Article Type: original scientific paper

Summary

The present study examined the difference in home affordances in young children with and without CHARGE syndrome. The results indicated that home affordances specially early affordances had a significant effect on fine and gross motor development in children with CHARGE syndrome rather than typical children. The results also indicated that children with CHARGE Syndrome current age had a significant positive relationship with a variety of stimulation, but not in children without CHARGE Syndrome.

Evaluation

The topic of this study is timely and interesting for publication in the IJERPH. The design for the study is appropriate to answer the research questions, and the paper is well written. For an original scientific paper, the manuscript is quite straightforward and I enjoy reading it. However, some points and suggestions should be addressed by the authors, in order to improve the quality of the manuscript.

Minor points and suggestions

  • Line 63, please delete the “status”, repeated twice.
  • Line 61-64. Please add the references for the sentence (These findings revealed a variety of factors that influenced the development of fine and gross motor skills at home including toy availability, parental physical activity involvement, parental socioeconomic status (SES), age, and level of children’s hear).
  • Please explain more about the logic of selecting the sample size as well as the sampling method in the participant’s section. for example, why didn't you use Gpower statistical software?
  • When was the study conducted? If it happened during the pandemic COVID-19, do not you think this situation can affect the response of families? It is better to write explanations in this regard as well.
  • How you calculated the age of each motor milestone for each child? please explain more in the manuscript (analysis section).
  • Why you used Mann-Whitney U tests and Spearman rank-order correlation? Did data have not normal distribution? please explain more in the analysis section.
  • Please add the effect size for each comparison. for example, Cohen’s d effect size for t-tests.
  • line 25-251. this is suggested to the author revise the sentence as follows: To the author's knowledge, this is the first study to examine the home environment in young children with a severe disability, such as CHARGE syndrome.

Author Response

Thank you for your comments.  We are pleased to hear that you enjoyed reading our paper. We address each of your specific in the attached file.
